# Monotherapy cancer drug-blind response prediction is limited to intraclass generalization

**William G. Herbert**[1,2,3,4]*, **Nicholas Chia**[5], **Paul A. Jensen**[6,7],
**Marina R. S. Walther-Antonio**[2,3,4]

1 Graduate School of Biomedical Sciences, Mayo Clinic, Rochester, Minnesota, United States of America, 2 Department of Obstetrics and Gynecology, Mayo Clinic, Rochester, Minnesota, United States of America, 3 Department of Surgery, Mayo Clinic, Rochester, Minnesota, United States of America, 4 Microbiome Program, Center for Individualized Medicine, Mayo Clinic, Rochester, Minnesota, United States of America, 5 Computing, Environment, and Life Sciences, Argonne National Laboratory, Lemont, Illinois, United States of America, 6 Department of Biomedical Engineering, University of Michigan, Ann Arbor, Michigan, United States of America, 7 Department of Chemical Engineering, University of Michigan, Ann Arbor, Michigan, United States of America

* willherb@umich.edu

## Abstract

Monotherapy cancer drug response prediction (DRP) models predict the response of a cell line to a given drug. Analyzing these models' performance includes assessing their ability to predict the response of cell lines to new drugs, i.e., drugs that are not in the training set. Drug-blind prediction displays greatly diminished performance or outright failure across a wide range of model architectures and different large pharmacogenomic datasets. Drug-blind failure is hypothesized to be caused by the relatively limited set of drugs present in these datasets. The time and cost associated with further cell line experiments is significant, and it is impossible to predict beforehand how much data would be enough to overcome drug-blind failure. We must first define how current data contributes to drug-blind failure before attempting to remedy drug-blind failure with further data collection. In this work, we quantify the extent to which drug-blind generalizability relies on mechanistic overlap of drugs between training and testing splits. We first identify that the majority of mixed set DRP model performance can be attributed to drug overfitting, likely inhibiting generalization and preventing accurate analysis. Then, by specifically probing the drug-blind ability of models, we reveal the sources of generalizable drug features are confined to shared mechanisms of action and related pathways. Furthermore, we observed that, for certain mechanisms, we can significantly improve performance by limiting the training of models to a single mechanism compared to training on all drugs simultaneously. Across multiple different model architectures examined in this paper, we observe that drug-blind performance is a poor benchmark for DRP as it does not describe model behavior, it describes dataset behavior. Our investigation displays that these deep learning models trained on large, monotherapy cell line panels can more accurately

**Data availability statement:** There are no primary data in the paper. The code used for all data preprocessing and experiments in this study is provided at https://github.com/willherbert27/drug_blind_generalization.

**Funding:** This work was supported by the Mayo Foundation for Medical Education and Research (WGH). The funders had no role in study design, data collection and analysis, decision to publish, or preparation of the manuscript. Salary for WGH was provided by the Mayo Foundation for Medical Education and Research.

**Competing interests:** The authors have declared that no competing interests exist.

describe mechanism of action of drugs rather than their advertised connection to broader cancer biology.

## Author summary

In this paper, we characterize the feature space of cancer drug-blind prediction. To understand the efficacy of a novel cancer drug it has never seen before (drug-blind), a model must be able to accurately compare this drug to all drugs it saw during training. These relationships between cancer drugs, the feature space, must be described well enough that this is possible. We believe that these relationships are poorly defined because cancer DRP models always display reduced performance in a drug-blind context. For the first time, we quantified the limits of generalization in a drug-blind setting. We showed that drug-blind generalization describes mechanistic relationships among drugs during model training. We also outlined new criteria with which to judge the drug-blind ability of a model. Failure of drug-blind prediction is an oft overlooked shortcoming in cancer DRP with potentially damaging downstream implications. We hope to show drug-blind ability of these models in a new light to guide others towards more pertinent tasks in cancer deep learning.

## Introduction

Five year cancer survival rates have almost doubled in the past five decades [1]. This a remarkable sign of our improved understanding of the disease. An inherent side effect of this is that we are left with increasingly complex cases that require an intricate examination of cancer biology [2]. Artificial intelligence (AI) has been touted as the solution to unlocking a more sophisticated understanding of cancer [3,4]. The translational function DRP models seek to recapitulate is the identification of a compound that will be effective in treating a specific type of cancer.

Over 100 different cancer DRP models have been published utilizing large pharmacogenomic datasets [4]. These models attempt to translate advances in AI to improve our understanding of cancer treatment. Recent work has focused on ensuring the robustness of these models by developing frameworks for reusability and standardized comparison [5,6]. While this increased rigor is useful for quantifying relative performance gains, the question remains: What are we actually learning about cancer treatment when training these models?

To answer this question, we can look toward methods for training and quantifying performance of cancer DRP models. Architecture-focused approaches hypothesize that different deep learning model designs will extract different, and perhaps better, information from cancer cell line drug response data. The majority of these models are trained on the same datasets and response metrics [4]. Most new models compare to out-of-the-box machine learning baselines rather than previously published

work [6]. Model reusability and concordance with published results is also low due to minimal detail on essential steps such as pre-processing or hyperparameter optimization [6]. This creates challenges in DRP model benchmarking.

Rather than designing different model architectures, we can examine how training data drives DRP model behavior. There are a variety of different large monotherapy pharmacogenomic cancer cell line panels used in cancer DRP training [7]. Early work examined the general scaling laws applicable to these datasets [8]. Partin et al. find that deeper networks achieve improved performance over gradient boosting methods that is more appreciable as dataset size increases. This is important for guiding further collection and experimentation. Different datasets also display different levels of efficacy for prediction among one another, showing that not all cancer DRP datasets are created equal [9]. Xia et al. show that models trained on CTRP can generalize well to other datasets, but this can be limited by experimental consistency, e.g., the assays used to measure response. They also identify that model performance responds more to increasing the number of unique drugs present in the dataset rather than the number of unique cell lines. They posit that increased information about mechanistic classes of drugs may underly this. These works set the stage for examination of cancer drug response generalizability.

We can evaluate model performance itself in different data-centric manners. The most common form of analysis is in the form of mixed-set testing, where all drugs and cell lines may appear repeatedly in training and testing. This is the least robust form of model evaluation. More translatable analyses are cancer-blind and drug-blind splits. These are models where the sets of unique cell lines and drugs, respectively, are disjoint in the training and test set.

Cancer-blind and drug-blind analysis both describe the ability of a model to generalize. Models are traditionally capable of cancer-blind prediction, but performance worsens significantly in a drug-blind setting and some models fail completely [4]. This failure is incredibly concerning; drug features should be generalizable in order to be transferable to newly identified treatments. The relevance of cancer cell lines themselves with respect to patients itself is already questionable [10]. If we are to determine the difference in drug generalization in a patient setting versus a cell line setting, we must first begin to understand drug generalization in a cell line setting alone.

It is important to thoroughly probe potential sources of failure when determining the capability of an AI model or dataset to perform a task [11,12]. This is particularly true when the downstream applications are sensitive, such as in healthcare. The shortcomings of large pharmacogenomic datasets themselves are well documented [13–16], but little has been done to explain causes of drug-blind failure. Our primary objective in this paper is to examine exactly what occurs during model training to drive drug-blind failure.

It is essential to place our work in the context of prior studies that perform drug-focused analysis of cancer DRP performance. Model performance has been shown to improve when trained on all drugs at once rather than on each drug individually. [17]. This work indicates that there is beneficial drug-to-drug information which appears to cluster by mechanism of action. It is also clear that there is disparate information that can be learned from each drug and that some drugs are more difficult to learn than others [18]. Taken together with the prior study, there appears to be both constructive information among drugs tempered by a heterogeneity among learnable features. It was found that experimental noise can damage learned chemical features, but binary representation of drug response restored some performance in drug-blind models [19]. Furthermore, specific aggregations of results can be used to provide a more reliable evaluation of drug-blind performance [20]. In one of the only efforts to directly address drug-blind failure, performance was improved using multi-objective optimization to prevent training from skewing performance to specific drugs by enhancing the extraction of global drug response features [21]. These global response features display that it is possible to prioritize drug-blind performance over individual drug performance. The features contributing to drug-blind and mixed-set performance may even be somewhat disjoint.

In this work, we sought to understand the specifics of what makes drug-blind prediction difficult. What this amounted to was the careful examination of how designing different training and testing splits impacts what a model learns and generalizes to. Previous approaches in drug discovery attempt to "learn" the optimal composition of training data to maximize

performance with as few training examples as possible [22]. There are also approaches in quantitative structure activity relationship modeling for rational selection of train/test splits [23]. In cancer drug response prediction, the overlap between chemical structures in the training and testing set presents a difficult confounding factor when analyzing model performance. Indeed, prior works examining drug-blind prediction discuss this as a limitation, even recommending that one might consider tuning the training and testing sets to particular model applications [20,21].

Rather than developing an application for addressing drug-blind failure, we sought to fully characterize drug-blind limitations. We find that, as hypothesized, drug-blind failure occurs because the feature space describing cancer drugs is incomplete. While individual mechanisms are well defined, global features describing cancer drugs are non-existent. We did not identify a model in which relevant information exists for a previously unseen drug if it does not fall into these well defined mechanisms. These different model architectures struggle to solve drug-blind failure because this information does not exist to be extracted from the actual data they are trained on. Moreover, it is difficult to purposefully fill in these gaps in the feature space. Further characterization can reveal relationships among pathways, but it does not appear feasible to predict these relationships in entirely novel mechanisms. Mechanism focused model training provides a potential solution by focusing learning on well-defined regions of the feature space. We conclude that future DRP work must prioritize appropriate depth and variety in a mechanism specific setting rather than breadth across global drug response.

The key contributions of this work are as follows:

1. We quantify how information sharing occurs across drugs in a drug-blind setting. We find that information sharing occurs primarily among drugs with the same mechanism of action and, to a lesser extent, drugs with related mechanisms. Certain kinase inhibitors with broader affinity benefit from shared information across many classes. These observations extend to advanced cell line representations as well as different model architectures.

2. We show that our tested models identify drugs with similar mechanisms during training using only cell line responses. These models reinforce mechanistic information, if it is available, rather than overfitting to individual drugs.

3. We observe confining model training to specific mechanisms of action better resolves individual drug-cell line pairs by decreasing drug to drug similarity. These models can learn behavior of drugs on specific cell lines rather than collapsing to learning mechanims of action.

## Results

### Permutation of response values reveals dependence on drug distributions

Drug response values were permuted either within drug (intradrug shuffle) or within cell lines (intracell shuffle)(Fig 1). This measures whether models are learning a true drug-cell line connection. Intradrug shuffling and intracell shuffling identify the dependence of a model on the specific pairing of a drug with a cell line or a cell line with a drug, respectively. We hypothesized that any impacts of permutation on performance are driven by the heterogeneity and distribution of learned responses rather than to learned label noise as in prior studies [13,17]. One-hot encoding removes all biological information and replaces it with a binary for the cell line present. This creates a similar disconnect from cell line to drug as intradrug shuffling, but retains true cell-drug pairings.

We expected that randomly shuffling the data would remove information about specific cell-line/drug pairings and reduce performance. Intradrug shuffling decreased performance by only 10–15% for all metrics across all three datasets, while intracell shuffling deteriorated performance by 60–90%. This suggests learning the overall response distribution of a particular drug, not any specific drug-biology connections, determines up to 90% of DRP model performance. Learning specific drug-biology connections is essential to exrapolating to unseen drugs and biological measurements taken from tumors; however, our models are overfitting to individual drugs without concern for cell line features. We repeated these

PLOS Computational Biology

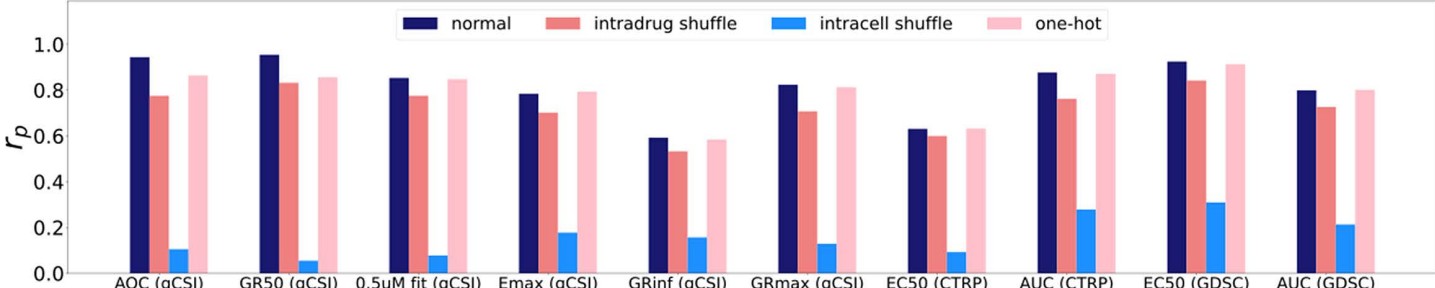

**Fig 1. Permutation of response data across cell lines and drugs.** We examine model performance across all included datasets and metrics for 3 different input conditions and a control. Response metrics are denoted on the x-axis, with source datasets included in parentheses. AOC: Area over the growth rate response curve. GR50: Concentration of drug at which growth rate is 50% of maximum response. 0.5uM fit: Growth rate at 0.5uM of tested drug. Emax: Effect of drug on cell viability at highest tested concentration. GRinf: Growth rate extrapolated from asymptote of the growth response curve. GRmax: Growth rate at the highest tested concentration of a drug. EC50: Concentration of drug at which cellular viability is at 50% of maximum response. AUC: Area under the dose response curve.

experiments using an advanced cell line representation (scFoundation) and a graph neural network architecture (Graph-CDR) to determine if this phenomenon extends beyond the simple MLP model we tested above [24,25]. We find similar patterns of performance decreases (Fig A in S1 Text).

A similar experiment that has been performed in previous DRP studies replaces a drug's response values with the average response for that drug [19,26]. This approach also effectively isolates performance to the information learned only from drugs. Both studies display only small decreases in performance in the drug-average benchmark. Li et al. in particular suggest that this approach should be used as the true baseline for DRP model performance rather than the minimum of whatever metric is being measured, e.g., a Pearson correlation of zero. We observed that, in intradrug permutation experiments, predicted response values tended to collapse to a constant value that corresponds to the mean response for that drug (Fig B in S1 Text). Therefore, using the incorrect cell line information forces model behavior similar to the "average response" approaches in Branson et al. and Li et al. The observed behavior of predictions collapsing to a constant value can also be used to diagnose poor quality or mismatched cell line information.

### Quantifying the impact of dataset diversity on performance

We next sought to examine the limits of learning the underlying distributions of drug response in the GDSC dataset using the EC50 response metric. All further experiments in this paper use GDSC EC50 as the training target. In particular, we examined the impact on performance of limiting the amount of cell lines that describe a drug's behavior; we refer to this as a dataset's diversity. We predicted that as a drug is tested on more cell lines, the better the drug's overall response distribution will be defined. In actuality, we saw there is no appreciable drop off until around 40 cell line examples (Fig 2A). Overall, this indicates that efforts to broaden the scope of cancer drug response datasets should focus on the addition of unique drugs rather than cell lines. This result confirmed prior work examining the utility of increasing the scope of cell line and drug experiments across datasets [9].

Interestingly, the variance of results obtained increased drastically between mixed set and drug-blind testing. Previous work has hypothesized that increasing the amount of unique drugs present during training would improve drug-blind performance [4]. We show empirically this does not occur on the scale of current large pharmacogenomic datasets. We also observe that different mechanisms of action are more sensitive to decreasing number of cell line examples (Fig 2B). This indicates that cell line specific information is used more by some mechanisms than others.

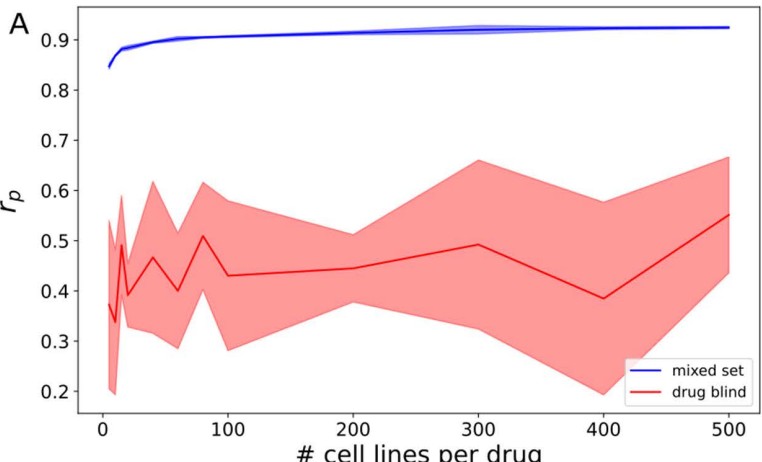
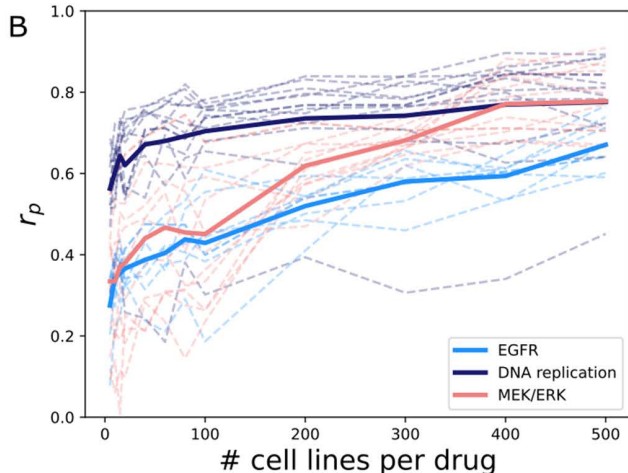

**Fig 2. Impact of varying the number of cell lines a drug is tested on. (A)** DRP model performance when increasing the number of cell lines per drug between mixed set and drug blind conditions. Shaded regions indicate standard deviation across five replicates with varying train/val/test splits. Increasing the number of cell lines per drug in the training set does not improve performance, but variance greatly increases from mixed set to drug blind testing. **(B)** Mixed set performance for specific drug mechanisms of action. Dashed lines represent individual drugs while solid lines are the average across all drugs in a mechanism. Specific mechanisms of action show greater sensitivity to decreasing numbers of cell line examples, indicating that cell line specific information matters more to some drugs than others.

## Training set composition predicts drug-blind performance

We next wanted to determine the association between the drugs contained in the training set and drug-blind performance. We do this by randomly varying drugs present during training across a set of models while holding the test set constant (Fig 3A). The Pearson correlation for all models trained varies between 0.517 and 0.169 with a standard deviation of .079. We fit an elastic net model with the binary presence of input drugs as features and model performance as targets (Fig 3B). The elastic net model showed a clear connection between drugs in the training set and overall performance. Accurate prediction is possible even with a small sample size of experiments. We further examine if performance is correlated with training set uniqueness (Fig C in S1 Text). Uniqueness in this case refers to the number of unique drug targets present in the training set. We find that this does not correlate with model performance, indicating that individual drug-to-drug information sharing is more responsible for the predictability of performance.

As a higher proportion of total drugs are included during training, overall performance variance will decrease. This is caused by the total potential overlap between training and testing sets approaching its upper limit. Because of this, creating a predictive model of performance requires that a smaller subset of total unique drugs be used. This is the reason for using a training set comprised of a random selection of exactly half of the unique drugs in the dataset.

## Drug-blind generalization is limited to functional relationships

We then extended the experiment from the prior section to an approach where training set composition is similarly varied, but the test set is composed of all drugs available in the dataset. Therefore, drugs are drug-blind in some models but not in others. An elastic net model was trained with binary presence of input drugs in the training set as the features and accuracy for an individual drug across all experiments as the targets. We consequently train an individual model for each drug. Coefficients for predictor drugs (drugs in the training set) are hierarchically clustered to create a heatmap of these associations (Fig 4). More positive coefficients indicate that the presence of a particular drug in the training set improves our prediction of response accuracy of a particular drug in the test set. Negative coefficients indicate a loss of predictability.

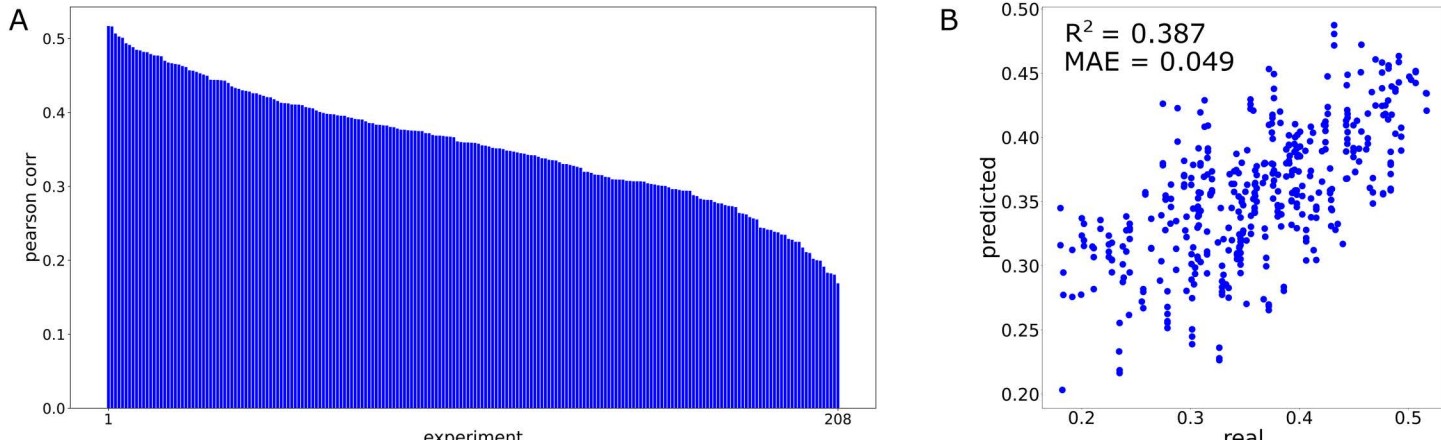

**Fig 3. Drugs in training set predict overall performance of drug-blind test set. (A)** Individual performance across 208 drug-blind models with varying training sets and constant test set. Each bar is a single model. **(B)** Predictions made by elastic net model fit to model performance using training set composition. Each point is a single model.

As expected, the diagonal of the coefficient heatmap contains the largest values, i.e., the best predictor for the performance of a drug in the test set is itself (Fig 4). This is the non-drug-blind case. For 224 out of 230 drugs (97%), the strongest predictor of performance is the non-drug-blind case. This gives some insight into the extreme performance drop in drug-blind experiments. During mixed set testing, drugs are highly dependent on their own examples for prediction. Furthermore, we see that coefficients in the center of the heatmap are incredibly sparse (Fig 4). Not only is the strongest predictor of performance almost always the drug itself, most drugs don't have any other predictors of performance. There exists little to no relevant information learned by the model being shared from drug to drug for the majority of drugs.

Outside of self-association, we see squares of shared coefficients. When we zoom in on them, these clusters reveal themselves to represent different cancer drug mechanisms of action (Fig 4B). The bulk of identified clusters are part of the RAF/MEK/ERK and PI3K/AKT/mTOR pathways. These pathways are central to cell proliferation and are both commonly mutated and targeted in many different cancers [27,28]. Interestingly, the coefficients determining drug predictive power extend across these pathways, which we term class-to-class relationships (Fig 5C). The presence of an ERK inhibitor in the training set improves the prediction of the MEK inhibitors in the test set. ERK is commonly represented as being immediately downstream from MEK in cellular signaling pathways. MEK is also overrepresented in the class-to-class relationships. It is coupled with microtubule destabilisers, DNA crosslinkers, dsDNA break inducers, and RNA polymerases. MEK is at the very top of signaling pathways controlling proliferation and cell surivival. mTOR inhibitors in the training set are also strong predictors for performance of PI3K and AKT inhibitors' accuracy. This predictive power is even stronger than in-class relationships between mTOR inhibitors. mTOR/PI3K/AKT are strongly interrelated, so identifying where one may influence the other can be informative when making a therapeutic choice [29]. Clustering also picks out a class of anti-mitotic drugs. The anti-mitotic drug relationships are granular enough that they separate taxanes (Docetaxel and Paclitaxel) from vinca alkaloids (Vinblastine and Vinorelbine) (Fig 4B). Anti-mitotic drugs have a class-to-class relationship with both MEK and ERK, which makes sense given the role they play in regulation of proliferation. We also can identify a family of bromodomain (BRD) inhibitors that are closely associated with one another but share almost no other predictive power with drugs outside this class. Finally, we identify a cluster of Top1/2 inhibitors. This cluster also contains Gemcitabine and Talazoparib. While both drugs do not specifically act on Top1/2, Talazoparib is a PARP inhibitor and Gemcitabine blocks DNA synthesis by mimicking deoxycytidine triphosphate. The GDSC supplied annotation identifies all of

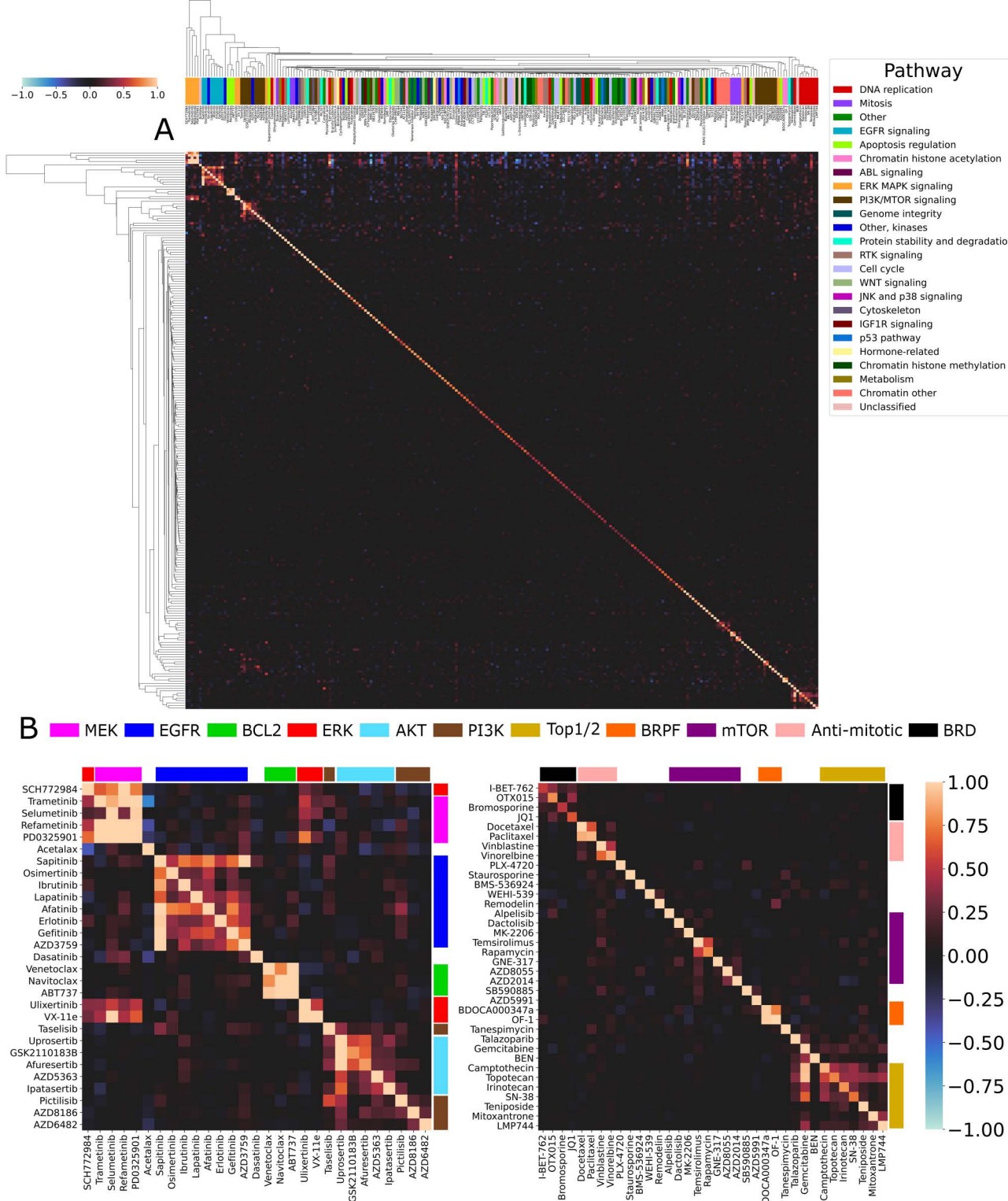

**Fig 4. Clustering of coefficients reveals intraclass drug-blind generalization. (A)** Drugs as features (y-axis) are trained to predict individual test set drug performance (x-axis) using elastic net across 1,641 trained DRP models. Dendrogram is created using agglomerative clustering of coefficients for each individual drug. The presence of the diagonal in this clustering dendrogram displays that most predictive power for a drug's accuracy is when itself is in the training set (non-drug-blind). Performance prediction outside of self-association corresponds to mechanisms of action. Cluster color indicates

the GDSC supplied pathway annotation. Plotted values are capped at [-1 1] to increase contrast and improve visualization. Best viewed zoomed in. **(B)** Magnified view of core cancer drug target pathways from (A) identified by drug-blind generalization. Coefficients from elastic net model fit to drugs as predictors (y-axis) of the performance of targets in the test set (x-axis). Drugs are color coded according to their advertised specific mechanism of action rather than GDSC supplied pathway annotations.

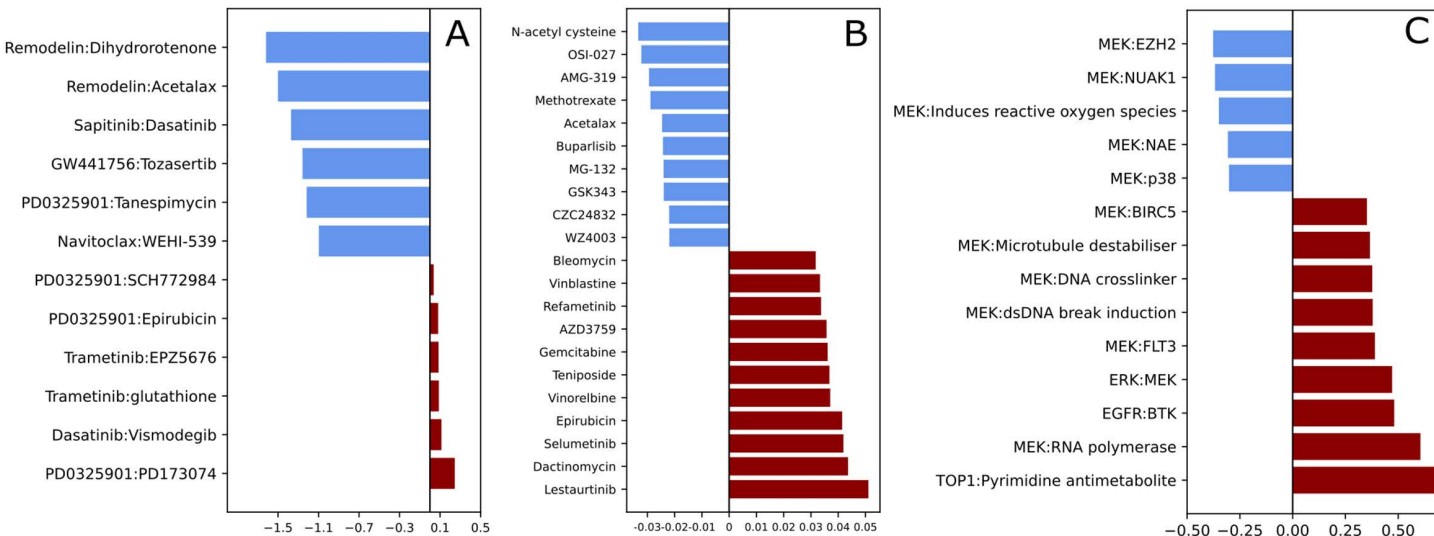

**Fig 5. Quantification of predictive relationships across cancer drugs in drug-blind models.** Coefficients for all following calculations are drawn from the heatmap in Fig 4A. **(A)** One-to-one drug relationships (predictor:target), calculated as the logarithm of the ratio of maximum non-self predictive coefficient to self-prediction. **(B)** Many-to-one drug relationships, calculated as the average of the sum of all non-self predictive coefficients. **(C)** Class-to-class (predictor:target) relationships, calculated as the average of interclass predictive coefficients.

these drugs as belonging to the "DNA replication" pathway. This relationship displays that intraclass generalization refers more to a general mechanism of action than the specific target of a drug.

We identified drugs with many different predictors of performance by summing the coefficients of their predictors and averaging by the number of unique drugs in the dataset (Fig 5B). These drugs can be identified by the vertical streaks of coefficients along the top and bottom of the heatmap in alignment with the identified drug clusters (Fig 4A). All RAF/MEK/ERK and PI3K/AKT/mTOR drugs generalized to prediction of Lestaurtinib, a tyrosine kinase inhibitor with demonstrated target promiscuity [30]. Lestaurtinib had the strongest many-to-one relationship of all drugs in this dataset. We also saw strong many-to-one relationships for the chemotherapy drugs Bleomycin, actinomycin D, and Epirubicin. Additionally, the center of the heatmap is sparse with no strong associations outside of the diagonal. This, along with the visualization of many-to-one relationships, show there is only a small set of drugs with the ability to generalize. All of the drugs with strong potential for generalization belong to core MEK/ERK or PI3K/AKT/mTOR pathways. It is important to note that there exist no drugs with global generalization ability. These drugs, if they existed, would appear as horizontal lines in the heatmap.

More difficult to visualize in the form of a heatmap are drugs that have a specific, uniquely strong predictor of performance (Fig 5A). We obtain one-to-one relationships by dividing the maximal drug-blind predictor of performance with the self-predictive coefficient and take the logarithm of this value. Drugs with values greater than zero are those that have a stronger predictor than themselves. There are only 6 such one-to-one relationships, with two drugs appearing across 5 of them - PD0325901 and Trametinib. Both of these drugs are MEK inhibitors. Interestingly, the drug for which PD0325901 exhibits strongest predictive power for performance is PD173074. PD173074 is a FGFR1 inhibitor, but also has been

shown to inhibit the MAPK pathway [31]. Such one-to-one relationships identified from predictive performance in drug-blind models can perhaps guide experimentation routes for poorly understood drugs.

It is also important to address the presence of negative coefficients in Fig 4. Negative coefficients indicate that a drug present in the training set makes us worse at predicting responses for a target drug in the test set. In one-to-one and many-to-one relationships, highly negative coefficients could indicate that drugs are structurally similar but have different downstream effects. This would make them difficult to separate during embedding of Morgan fingerprints. We can see an example of this in Fig 5B. CZC24832 and AMG-319 are both PI3K inhibitors, but they do not cluster together with other PI3K inhibitors and display some of the most negative many-to-one relationships in the whole dataset. CZC24832 specifically inhibits PI3K$\gamma$ while AMG-319 inhibits PI3K$\delta$ . The isoforms of PI3K display distinctive properties, both in benign cells and in cancer, and the drugs used to specifically target them have distinct structures [32,33]. Interestingly, other PI3K drugs do not contribute negative coefficients to AMG-319 or CZC24832. Instead, it is mostly the presence of MEK inhibitors in the training set that drive their poor drug-blind performance. Ideally, the learning of one drug should not negatively impact the learning of another. Mitigation of negative predictive power in specific situations such as the PI3K isoforms should be a goal of future work during DRP model development.

## The impact of alternative cell line representation and model architectures on drug-to-drug information sharing

It is valuable to determine the extent to which different cell line representations and model architectures contribute to drug-to-drug information sharing. Our method for determining this level of information sharing allows us to directly measure their impact in a drug-blind setting. Different cell line representations drive differing levels of performance in DRP models, with multi-omic integration showing gains over any single representation [34]. Recent single-cell foundation models benchmark their cellular representations against DRP and drug-blind prediction tasks [24,35]. We sought to determine if we could identify sources of these drug-blind performance gains within drug information sharing.

Using scFoundation embeddings results in similar mechanistic clustering of drugs to using unfiltered gene expression information (Fig D in S1 Text). The same primary groupings of GDSC target annotations reappear – MEK/ERK, DNA replication, PI3K/mTOR, EGFR, and Mitosis. Importantly, there is far less information sharing occuring outside of intraclass relationships. This indicates that there is information in unfiltered gene expression data that allows information sharing to drugs with broader mechanisms of action, such as Lestaurtinib. Embeddings from scFoundation are trained on single cell data, but they were implemented in a DRP setting with bulk cell line expression [24]. This potentially indicates that there is information relevant to drug-blind generalization lost in this process. Even within intraclass relationships, we can visualize less information sharing than in our base experiment (Fig 4).

If different models are truly able to extract different information during drug-blind testing, then our method will allow us to identify if there are specific architectures we should prioritize for drug-blind methods. We examine this behavior in two different graph-based architectures from the XGDP model [36]. XGDP uses GDSC1, while our original experiments use GDSC2. Thus, there is only a partial overlap of drugs. The first model we examine is a normal graph neural network architecture (Fig E in S1 Text).

We see strong clustering of MEK/ERK and EGFR classes, but we lose definition of the broader mitosis and DNA replication classes. Vinorelbine and vinblastine – both vinca alkaloids – are now separated from one another. DNA replication drugs Mitomycin-C, Gemcitabine, and Doxorubicin now appear in a more heterogenous information sharing group. We also see the appearance of new clusters. The grouping of 'Other, kinases' Saracatinib, A-770041, WH-4-023, and Dasatinib is not present in our prior experiments. Upon further review of these drugs' targets, they are all Src inhibitors. Dasatinib and Saracatinib also both inhibit Bcr/Abl. In the original experiment, though, our training data only contained the drug Dasatinib. When more drugs that share its mechanism of action are added, more information sharing occurs.

We also observe a new cluster of 'Protein stability and degradation' drugs and 'Chromatin histone acetylation' drugs. Again, most of these drugs do not appear in our original training set. Interestingly, they cluster with the drug CUDC-101,

which inhibits histone deacetylase, but is labelled as 'Other' by GDSC as it also inhibits EGFR and ERBB2. CUDC-101 shares little information with EGFR inhibitors, though, and primarily clusters with these chromatin histone acetylation targeting drugs.

There are also distinct negative coefficient streaks in the drugs DMOG and AICA ribonucleotide. This is a continuation of the strong negative information sharing of metabolism drug N-acetyl cysteine in our original experiment (Fig 5). The presence of many cancer drugs during training actually damages our ability to predict the response of metabolism targeting drugs in the test set. 'Selfish' metabolic reprogramming is a hallmark of cancer cells, known as the Warburg effect, and this is thought to drive most of the increased proliferation that MEK and ERK drugs target [37]. This would indicate that, at the very least, there would not be negative information sharing between metabolism and proliferation targeting drugs. When we examine the similarity of Morgan fingerprints between each metabolism drug and a set of MEK inhibitors (Refemetinib, Selumetinib, Trametinib, PD0325901, and CI-1040), DMOG, N-acetyl cysteine, and AICAR have average cosine similarities of 0.809, 0.814, and 0.812 respectively. These similarities are higher even than the average intraclass cosine similarity of the above MEK inhibitors, which is 0.531. If the fingerprint representation of these drugs is similar, but they have very disparate efficacy in the same cell lines, then the model will be unable to separate them without sufficient data. Indeed, we observe that MEK inhibitors and metabolism drugs have distinct response profiles in leukemia and melanoma cell lines (Fig G in S1 Text). It should be noted that XGDP uses a graph representation for drug structure rather than Morgan fingerprint, but this is still unable to overcome structural similarities that drive negative information transfer from MEK/ERK drugs to metabolism drugs. Overall, this indicates a need to find more effective drug representations for DRP to avoid potential sources of negative drug-to-drug information sharing and improve drug-blind generalization.

We then performed the heatmap experiment for a graph attention network (GAT) in XGDP (Fig F in S1 Text). We observe similar clustering of broader classes as the graph neural network. One notable exception is in the case of PI3K/mTOR drugs. There is much better definition of these classes in the GAT model. A PI3K inhibitor, Dactolisib, has one of the most negative many-to-one measurements in the GNN model. This does not continue when using a GAT model. Furthermore, the metabolism drug AICA Ribonucleotide appears as one of the drugs with the greatest many-to-one information sharing in GAT. This is also captured in the class-to-class relationship MEK:AMPK agonist. These two pieces of evidence may indicate that model architecture can indeed be used to mitigate negative information sharing across drugs. Another notable difference is that GNN has only one one-to-one pairing greater than 1, while GAT has eleven. This indicates that GAT is more likely to overcome non-drug-blind bias, albeit on a small scale. The two models do share similarities in their measured drug relationships as well. DMOG has the most negative many-to-one information sharing in both models, while Mitomycin-C is one of the highest. MEK, microtubule drugs, and DNA crosslinkers are also well represented in both models' class-to-class relationships. This further confirms the crosstalk between core cell proliferation pathways observed in the MLP model. We conclude that there are both positive and negative tradeoffs between GAT and GNN, and further application of this method to novel architectures can reveal improvements in more specific areas.

## Model training reinforces intraclass generalization in mixed-set testing

We examined the embeddings created from the drug arm during mixed-set testing to determine whether intraclass similarity is increased during model training. Using the supplied pathway annotations from the GDSC2 dataset, we plotted tSNE embeddings of a subset of drugs from the families that were identified during drug-blind testing. We saw that, before training, there was some clustering of EGFR signaling and DNA replication classes when just examining raw Morgan fingerprints, but most drugs were well separated from one another (Fig 6A). During training, we saw that by epoch 40, drugs separated into distinct clusters corresponding to mechanism (Fig 6C). This initial separation corresponds to the steepest region of loss during training (Fig 6B). What follows from epochs 40–100 appears to be more of a refinement period, where boundaries between drug mechanisms are reinforced. We observed a close relationship among MEK, EGFR, and some PI3K/mTOR drugs and another combined cluster of DNA replication and mitosis drugs. This reiterates the earlier

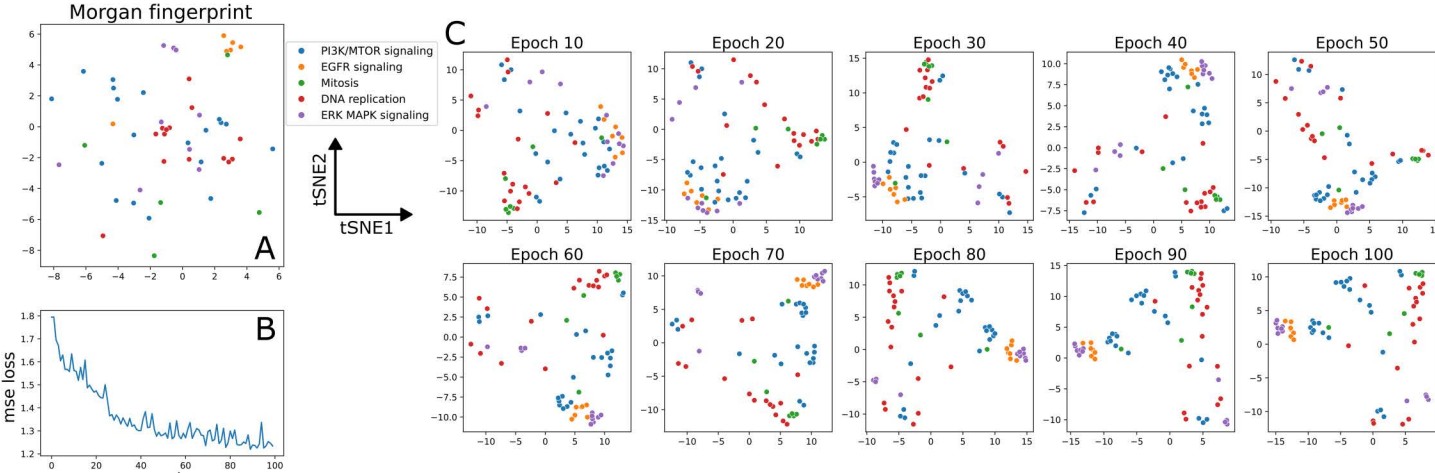

**Fig 6. Intraclass similarities of drug embeddings increase throughout training.** (A) tSNE representation of raw Morgan fingerprints for drugs with select pathways of action. There is no clear delineation between mechanisms in this representation. **(B)** Validation loss throughout training for drug response prediction model trained on GDSC to predict EC50. We see a steeper loss region from epochs 0 to 40 with more refinement occuring from epochs 40 to 100. (C) tSNE representation of drug embeddings at the last layer of the drug arm across 100 epochs of training. Borders between mechanisms become more defined throughout model training.

point that while drugs with a shared mechanism are most similar in the drug feature space, models are also aware of broader pathway relationships.

Some broader pathway features remain disperse. For example, when we examine the distinct regions occupied by PI3K/mTOR drugs in epoch 100 of Fig 6C, we see they split in to three primary groups. The first is composed of all AKT inhibitors and some PI3K inhibitors. This cluster is closest to the EGFR inhibitor cluster. The second is composed of a mixture of mTOR inhibitors and PI3K inhibitors. This is between the first PI3K/mTOR cluster and the DNA replication and Mitosis cluster. The third are outliers in the bottom right of the figure and they include the drugs with high negative information sharing discussed earlier - AMG319 and CZC24832. This indicates that AKT and mTOR inhibitors are defined as distinct from one another in the model. This also indicates that there are some PI3K drugs more similar to AKT inhibitors than mTOR and vice-versa within this pathway level label.

This approach also allows us to visualize how narrow the feature space occupied by each mechanism is. The features extracted by model training correspond strongly to overall mechanism of action. If a new mechanism were added to the dataset, such as in the case of a novel drug, there exists little information to correctly place it relative to all other drugs. Only once a mechanism is added and appropriately characterized would we be able to see the feature space it occupies.

## Training on specific drug mechanisms improves performance compared to training on all drugs at once

We next sought to determine if model performance could be improved by training on only one mechanism of action at a time. We hypothesized that models trained in this manner would be better able to resolve intraclass differences in response. Training was performed on the same five drug mechanisms to be well defined during model training. Comparisons were made to a whole dataset test instance containing the exact same mechanism specific cell line and drug pairings comprising the test sets of mechanism specific models.

In all tested mechanisms, performance improves when training on a single mechanism compared to training on all mechanisms at once (Fig 7A). We are able to achieve this performance using up to ten times less data than in whole model training. We expected that the primary source of this improvement was that the model is able to identify the specific

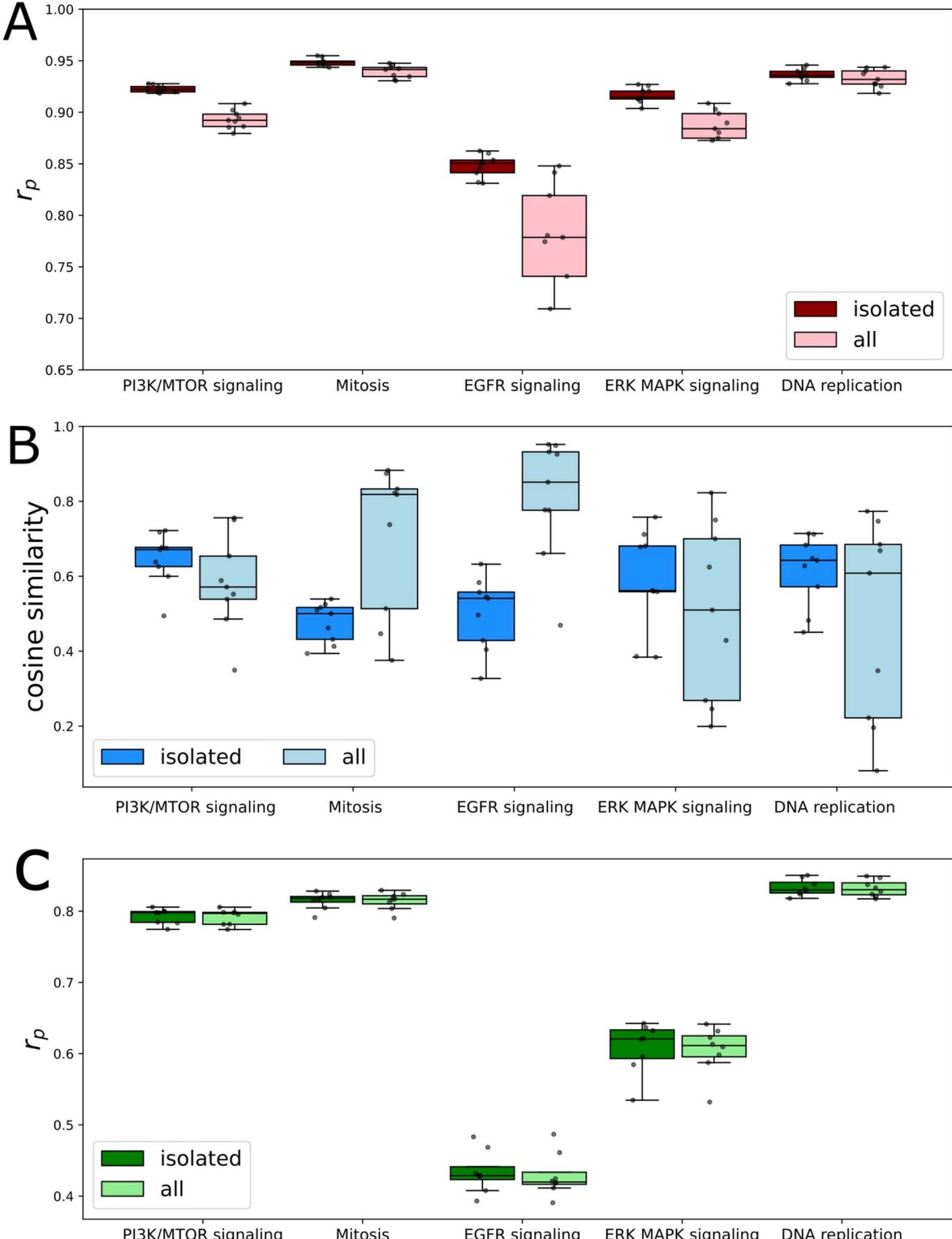

**Fig 7. Training on specific mechanisms improves performance by decreasing intraclass similarity.** Models are trained on a single mechanism (isolated) or trained on all drugs at once (all) to predict IC50 (GDSC). The test set when training on all drugs at once is created to intersect with the test set of all specific mechanisms. **(A)** Pearson correlation comparison in different mechanisms of action. **(B)** Average pairwise cosine similarity among all drugs calculated from embeddings at the last layer of the drug encoding arm of a fully trained model. **(C)** Intradrug permutation performance compared between mechanism specific and whole dataset trained models.

action of individual drugs rather than collapsing all drug representations to a mechanism. Comparing the average pairwise cosine similarity of drug embeddings in each mechanism, we indeed see that drug representations become more dissimilar in the mechanism specific models that benefit most from this type of training (Fig 7B). To ensure that models are not more closely fitting individual drug distributions, we repeat our earlier intradrug permutation experiment at a mechanism-specific level (Fig 7C). We can compare performance in Fig 7A to these values as a drug-average benchmark, as in [19] and [26]. We see that the drug permutation performance remains the same across all mechanisms, so increases in performance in normal training are increased relative to this baseline.

Decreased cosine similarity is not present in PI3K/mTOR, ERK/MAPK, or DNA replication specific models. This is likely due to the fact that these classes contain drugs with more variety in their specific targets. We further divide the PI3K/mTOR pathway into its constituent targets - PI3K, mTOR, and AKT – and train mechanism specific models on these groupings (Fig H in S1 Text). We see that improved performance and decreased cosine similarity are present in the mTOR and AKT pathways, but not in PI3K. This displays that further specificity may be required if annotated targets describe a pathway rather than a specific target. Furthermore, this improved split also aligns with the distribution of PI3K/mTOR drugs in epoch 100 of Fig 6C that was outlined earlier. mTOR and AKT drugs are distinct, with PI3K drugs spread between these groups and across other areas of the drug feature space.

## Discussion

In this work, we set out to identify quantifiable sources of drug-blind failure. Drug-blind failure is common in published DRP models. It is helpful to know whether this is an architecture or data failure. We primarily find that, across the models tested in this paper, it is a data failure. We displayed that across different datasets, response metrics, cell representations, model architectures, and prediction targets that a large source of learning is simply fitting to the distributions of drugs. We found that there is little global utility to increasing the number of cell line experiments for a particular drug, but certain mechanisms of action do respond differently to different levels of information. We also observed that knowledge of the drugs present during training allowed us to predict the performance of a model with high accuracy. We then measured the amount of drug-to-drug information sharing occurring in a drug-blind setting, and therein found the quantifiable measurement of drug-blind failure we wanted to define.

Information sharing in a drug-blind setting is limited to functional relationships between drugs, which we term 'intraclass' generalization. Neither advanced cell line representations or more advanced model architectures we tested were able to overcome this. In fact, when a different set of drugs is used for training, it reveals that drugs will receive and share information only if they have a sufficient number of 'intraclass' members present. We then showed that models will also collapse representations of drugs in to these classes in mixed-set testing. This, coupled with the fact that there is so little information sharing outside of mechanisms, caused us to hypothesize that models could be trained in a mechanism specific manner with little loss of information. We display that it is possible to increase performance in this manner, and that this performance is closely tied to a decrease in intraclass drug similarity calculated by the model.

The NCI60 team displayed that a neural network had the ability to predict drug mechanism of action based solely on cell line responses over 30 years ago [38]. Even given advanced representations of cell lines and expansion of pharmacogenomic datasets to hundreds of thousands of experiments, we have yet to test a model that does not collapse to representing mechanism of action in order to infer responses. Drug-blind failure simply describes something that has already been shown: cell line based drug response can be simplified to a function of drug mechanism of action. We believe that drug-blind testing should not be viewed as an appropriate benchmark for cancer drug response prediction models. Examining the behavior of models in a mechanism specific context is more pertinent to understanding drug generalization in the context of large pharmacogenomic datasets. This would involve training and testing models according to more specific use cases, such as a particular patient populations. Such a method was published very recently, specifically for PI3K$\alpha$ [39].

Drug-blind prediction is primarily relevant to identifying the efficacy of novel compounds. An ideal model with a global understanding of cancer drug generalizability would be adept at determining the effectiveness of any given drug in a zero-shot setting, not just one for whom it understands the mechanism of action. We found that we cannot expect this global generalizability in cancer drugs in the context of currently existing datasets and drugs, but there is still utility for examining understudied compounds. 63% of approvals of compounds for cancer treatment by the FDA from 2009 to 2020 were next-in-class, i.e., they used a previously described mechanism of action [40]. If drug-blind prediction is capable of intraclass generalization, then it is capable of generalizing to prediction of performance for the majority of newly approved compounds. In fact, mixed set training behaves inherently like contrastive learning among the known mechanisms of drugs (Fig 6C). Supervised contrastive learning can be used to train models that amplify features of intraclass similarity [41,42]. Cancer DRP models recreate this behavior without any class labels or high similarity of Morgan fingerprints, only by learning shared cell lines responses. Next-in-class prediction is therefore a realistic drug-blind task, especially with models tuned to specific mechanisms. A potential solution for prediction within new mechanisms of action would be to create models with built in uncertainty, such as in [43]. Given the sparsity of generalizability, drug-blind models should be aware of what they do not know given the information they already have to prevent inaccurate predictions. Drug-blind models could then mark drugs as poorly understood or even for further experimentation, such as in a lab automation setting.

There is an argument to be made that some representations of drugs in deep learning could be more generalizable than others. That is, these representations would better bridge the feature space among drug mechanisms of action. Among other studies that examine drug-blind prediction, drugs are represented using molecular graphs and graph neural networks [36,44–46], word embeddings [19,47], and canonical SMILES for convolution [48]. We refer readers to [4] for a more complete review of studies that utilize drug-blind benchmarks. Of the results that predict continuous drug response values and report Pearson correlation for comparison, none outperforms the distribution we observe in this paper (Fig 3). The results of this paper suggest that given the wide distribution of performance values, it is essential that an appropriately large amount of training splits are tested. This is because drug-blind analysis is so sensitive to the chosen train/test split due to changing mechanistic overlap. Only [48] performs a more appropriate amount of replicates, with 150, while others perform 3- and 5-fold cross validation. We recommend that future attempts to benchmark any improvement in drug-blind performance analyze a sufficient amount of replicates to cover the number of unique drugs in the dataset.

We observe drug-blind failure in a variety of model architectures. We reinforce this idea by testing our approach for quantifying drug-to-drug information sharing in an advanced cell line representation as well as multiple graph-based architectures (Figs D, E, and F in S1 Text). Furthermore, it appears that optimization based approaches trump architectural design when considering approaches to directly improve drug-blind generalizability [21]. In other words, changing how drugs share information is more important than changing what information is shared. As we observe in this paper, the amount of information that drugs across mechanistic classes are able to share is quite minimal regardless.

When identifying treatments for patients, physicians often determine candidate drugs using a particular mechanism of action that addresses a concrete event, such as a mutation. This, in theory, translates better to a classification task where we can determine whether a certain drug exhibits the desired mechanism of action. As we saw, this is a task DRP models are highly capable of. Unfortunately, empirically determining mechanism of action is a difficult task and results often do not belong to one discrete class [30,49]. As we have seen, the majority of drugs do not cluster into a well defined class in a drug-blind setting (Fig 4). Take, for example, the case of Lestaurtinib both in this paper and in [30]. The advertised target kinase of Lestaurtinib is not even in the top 20 targets for which it is actually the most active. Indeed, Klaeger et. al find that this is the case for many different kinase inhibitors. In the context of this promiscuous behavior of drugs, how do we identify effectiveness given particular cancer cell behavior? The answer to this question lies in the interpretation of many-to-one relationships in drug-blind prediction. The results in Fig 5B should not necessarily suggest that Lestaurtinib can function as a MEK, EGFR, or ERK inhibitor. Rather, the downstream effects of Lestaurtinib and these core pathways merge at some point. Since we know that Lestaurtinib can

exhibit such a broad array of bindings, it is more likely for its inhibitory potential to overlap with another given drug's at any time. For example, it has been found that Lestaurtinib exhibits synergistic effects through MAST1 mediated MEK activation [50]. Generalization outside of a specific mechanism of action is perhaps then more indicative of potential drug synergy. Recent work has shown that categorical embeddings of drug mechanisms can be used to enhance drug synergy prediction [51]. Rather than training categorical embeddings directly, embeddings from DRP models could perhaps be used as they encode mechanistic information while maintaining flexibility as in the case of Lestaurtinib. Drug synergy prediction is an entirely different, and potentially more clinically relevant, task in cancer DRP [52]. We are not recommending that we only perform classification of mechanism in drugs, but rather that our approach to analysis of drug-blind testing can better describe these drugs with more ambiguous activity. Determining drug-to-drug performance benefit is perhaps more helpful for understanding ambiguous drugs' responses than predicting some continuous response variable itself.

We finish with four recommendations to the broader cancer DRP community based on our results. The first is that all future DRP models should be benchmarked against a drug average or intradrug permutation benchmark in agreement of the results of this paper and Li et al. [26]. The 'worst-case' performance of a DRP model is the case where it uses zero information about cancer at all. The second is that the traditional drug-blind benchmark should be replaced with the drug-to-drug information sharing experiment outlined in this paper. Improved prediction for a single drug in a drug-blind setting is an incredibly noisy measurement. It is far more impactful to show that a model improves drug-blind generalization where there formerly was none than to display an 'improvement' in a small number of drugs. Furthermore, the required number of replicates to confidently display an improvement in drug-blind performance provides the required information to calculate information sharing anyways. Increasing the scope of models tested in this manner will further define our understanding of the limits of drug-to-drug information transfer. The third suggestion is that it essential to stratify model training and evaluation in a mechanism dependent manner. We believe that, even outside of cancer, mechanism specific evaluation represents a more accurate and clinically translational way to evaluate model generalizability.

The final suggestion is to prioritize measurement of how different experimental platforms for evaluating cancer drug response capture different levels of complexity. 2D liquid culture of cancer cell lines poorly mimics the heterogeneous physiological conditions of the human body, but that is the primary data source for DRP models. Our drug-to-drug information sharing method applied across different models – 3D organoids or PDXs in different animal models – could provide a true heuristic for the relevancy of these systems to patient treatment, but this is currently a data-limited problem. Such an approach aligns with the current goals of the U.S. Food and Drug Administration regarding a reduction in animal modeling [53]. If, for example, murine PDXs recapture the same level of drug-to-drug information sharing – and thereby complexity – as an organoid system in a particular cancer, this provides quantitative evidence for a reduction in animal testing requirements and also reducing the experimental burden of drug screenings in the process.

## Conclusion

Determining the ability of cancer DRP models to generalize to novel drugs is an important component for eventual translation to open-world clinical problems. This is traditionally performed by randomly splitting the unique drugs present in training and testing sets. In this paper, we quantify the extent to which the dataset itself confounds traditional drug-blind testing. We find that there exist distinct regions of information sharing, and these regions correspond directly to mechanistic overlap of drugs. A more accurate measurement of drug-blind ability in future models is to determine performance gains in these well defined regions. We have created a novel framework for determining DRP model generalization that is agnostic to dataset confounders. It is straightforward to implement with any model architecture as it only requires modification of train/test construction and sufficient training replicates. We hope that future DRP work can leverage this approach to explore the true upper limits of drug generalization.

## Materials and methods

### Drug response datasets

All cell lines were represented using gene expression data. Expression data was obtained from DepMap using Expression Public 24Q2 [54]. Cell lines without expression data from DepMap were filtered out from their respective datasets. We used DepMap as a constant source for gene expression data due to potential experimental differences across other collections and need to minimize sources of variance in model input. Gene expression features were filtered to intersect with patient expression data from the Cancer Genome Atlas (TCGA) [55].

Drugs were represented as Morgan fingerprints for input to our model. We first obtained the SMILES string for each drug using the PubChemPy (v1.0.4) python library. The PubChem database was queried using the supplied name of the drug from each dataset and the best match was used. Morgan fingerprints were generated using the GetMorganGenerator function of RDKit (Version 2025.03.6) with a radius size of 2 to return a fingerprint vector of length 2048. Drug SMILES were not manually annotated. If they were not automatically retrieved from PubChem, they were removed from the dataset. Reported dataset sizes below are after removal of cell lines without available gene expression data and drugs without an available PubChem SMILES.

1. **gCSI [56]** The Genentech Cell Line Screening Initiative was obtained from http://research-pub.gene.com/gCSI_GRvalues2019/. This dataset was of particular interest as it includes growth rate metrics (GR50, etc.) as an alternative to IC50. There were 43 unique drugs. Number of unique cell lines varied between 545 and 565.

2. **CTRPv2 [57,58]** Data from the Cancer Therapeutics Response Portal (v2) was obtained from the DepMap portal under the CTD2 release. There were 496 unique drugs and 839 unique cell lines present in the dataset. Total number of pairings after all preprocessing was 365,321 for AUC. CTRP communicates EC50 as 'apparent $\mu$M', so values were filtered out above $10^{14}$M and below $10^{-26}$M as these are extreme and disrupt training. There were 266,709 pairings for CTRP EC50.

3. **GDSC2 [59,60]** Data from Genomics for Drug Sensitivity were downloaded from https://www.cancerrxgene.org/downloads/bulk_download. We specifically utilized GDSC2. There were 230 unique drugs and 936 unique cell lines present in the dataset. Total number of pairings after all preprocessing was 194,491.

Initial permutation experiments were performed across all three datasets and all related metrics. All further experiments were performed on only GDSC2 using EC50. Mechanism of action was obtained from the compound annotations made available by GDSC.

### Model architecture and optimization

The deep learning model architecture used in this work is comprised of a multilayer perceptron (MLP) with two arms for separate embedding of drugs and cell lines (Fig 8). Cell line and drug representations are then concatenated and used to predict a continuous response value defined by the input dataset. Input to the model was therefore in the form of (cell line, drug, response) tuples. This simple model architecture was chosen in order to provide a constant baseline that was able to train rapidly with good performance. The layer dimensions of our model are chosen based on MLPs used for DRP dataset examination in prior work [8,9,61].

Unless otherwise specified, models were trained with a batch size of 16 using the Adam optimizer with a learning rate of 1e-4. These hyperparameters were chosen using a grid search over batch sizes [16,32,64,128,256] and learning rates [0.1, 0.01, 1e-3, 1e-4, 1e-5] in mixed-set testing. Models were trained using early stopping. A validation set was used to determine the epoch of early stopping and all performance is reported using a further test set. Data was divided into train/validation/test sets using an 80/10/10 splitting strategy. While we maintained a simple and constant model training strategy in order to more quickly and clearly examine the impact of data the model is trained on, we must recognize that more

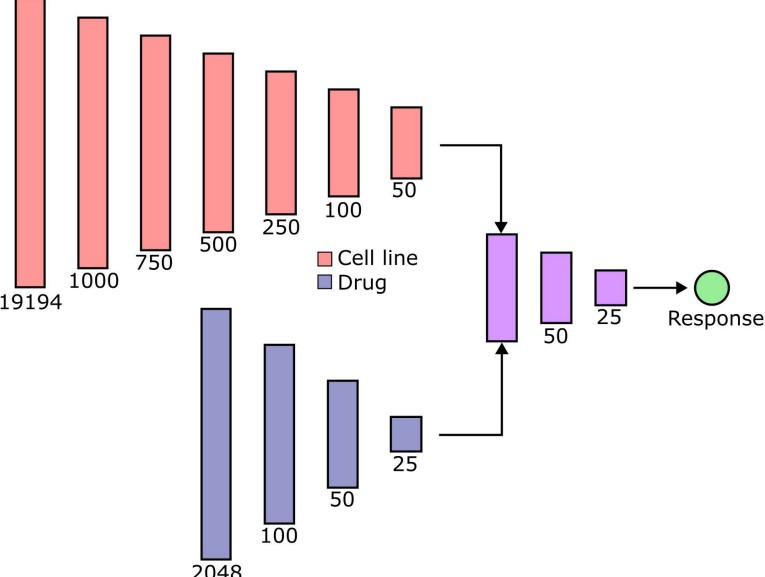

**Fig 8. Model architecture. The model used in this study is comprised of a multilayer perceptron with separate drug embedding (blue) and cell line embedding (red) arms.** Each embedding arm is comprised of successive linear layers with decreasing dimensionality. These embeddings are then concatenated to calculate a given response. All fully connected layers, excluding output, are followed by ReLU activation.

fine-grained hyperparameter tuning could potentially improve performance if one was interested in achieving state of the art drug-blind model ability.

## Permutation experiments

Intradrug model permutation was performed by randomly shuffling response values across all cell lines to which a particular drug was applied, as in [17], i.e., for a single drug, the distribution of response values went unchanged. The response values were assigned to a different cell line than the original. In theory, this should disconnect drug representations as a predictor of its response from using specific information about a cell line it is applied to. We also randomly permuted values across all drugs applied to a particular cell line, called intracell shuffling. In one-hot encoding, cell lines were represented with one-hot vectors rather than gene expression.

## Diversity of dataset composition

Dataset composition experiments were performed using the GDSC dataset. When testing the impact of the number of cell lines present for a given drug, the dataset was partitioned to only include drugs tested on 500 or more cell lines. This allowed the pool of drugs to remain the same across all diversity experiments. Filtering the drugs in this way reduced unique drugs from 230 to 214 drugs in GDSC2. The training set was then further filtered to only include the specified number of cell lines. Drug-blind experiments refer to those for which the drugs available in the training, validation, and test sets are disjoint from one another. When examining the impact of quantity of unique cell lines on mixed-set performance, the test set was held constant across all replicates. This was done in order to have a constant set for quantification of increased generalization. Five replicates were performed for each experiment in order to report standard deviation.

## Prediction of performance by training set composition

By varying the data present in the training set of a model, we sought to determine the impact of the inclusion of different drugs on model performance. For these experiments, we again utilize the GDSC2 dataset. The test and validation sets

across all experiments are held constant at 10% (23) of unique drugs while the training set is varied. Training set composition refers to the binary presence or absence of a drug in the training set. The total number of drugs in the training set at any given time was comprised of 50% (115) of the total unique drugs. To predict drug-blind performance based on training set composition, we trained an elastic net model [62] using scikit-learn v1.7.0 (alpha = 0.01,lambda = 0.1) using training set composition as features and accuracy as targets. Alpha and lambda values were chosen using the ElasticNetCV function multiple times across a subset of drugs with lambda (L1 ratio) values ranging from 0 to 1 with a step size of 0.01 and alpha values ranging from 1e-5–100 with power-10 step sizes. Features were binary vectors indicating the presence or absence of a particular drug in the training set and target values were the calculated Pearson correlation of that particular experiment.

To further elucidate the relationship between individual drugs, we created an experimental setup where the composition of drugs in the training set varies but the test set contains all 230 unique drugs in the dataset. In this way, the test set contains both drug-blind and non-drug-blind examples. We do this so that we have a constant set of targets with which to fit the elastic net model after DRP model training. The training set was again partitioned to include 50% (115) of the unique drugs present in the dataset. The validation and test sets contained the remaining non-overlapping cell lines for drugs present in the training set as well as all examples for drug-blind pairs. We trained 1,641 drug response prediction models total for these experiments. We then calculated the performance on a drug-by-drug basis for each DRP model using Pearson correlation. Because we can obtain individual drug accuracy for all drugs across all models, we can fit an elastic net model to the performance of each drug. Input features to the elastic net are the drugs present in the training set, binarized, and targets are per model Pearson correlation for the current drug. We summed and averaged the coefficients obtained for each drug across 10-fold cross validation. Using these coefficients, we performed hierarchical clustering on the predictive ability of each drug in the training set. We chose elastic net because our analysis was dependent on accurate measurement of all drug-to-drug relationships. By combining L1 and L2 regularization, elastic net is able to perform strong feature selection while avoiding the pitfall of only choosing one input feature out of a group of highly correlated features. This was particularly important since we wanted to define relationships among drugs with similar mechanisms. It was also essential that the chosen model not collapse all features outside the non-drug-blind case to zero because that is such a strong predictor.

We repeated the partially drug-blind experiment in 3 alternative settings to determine if cell line representation or model architecture contributed to different levels of drug-to-drug information sharing. The first repeat used scFoundation cell line embeddings [24]. scFoundation is a foundation model trained using single-cell gene expression data. The original publication uses embeddings from this single-cell model to perform cancer DRP on bulk cell line gene expression data. The scFoundation embeddings contain 768 features. We therefore removed the first two linear layers (1000 and 750) of the model architecture and input these embeddings from the third linear layer (500). All scFoundation experiments were performed with the same training code as our original experiments. The further two experiments utilized graph neural networks from XGDP [36]. The original XGDP manuscript compared the utility of different graph-based architectures for generating explainable drug features, and they provide a useful framework for comparing these architectures. The only change we made to the code of XGDP was an edit to the train and test set construction to implement our partially drug-blind approach. XGDP uses the GDSC1 dataset, while we use GDSC2, which means that our drug predictions are not fully overlapping. We specifically chose to benchmark their graph neural network (GNN) and graph attention network (GAT) implementations. scFoundation and XGDP experiments were both trained until a set of 1600 models with unique training set compositions was reached.

## Supporting information

**S1 Text. Supplementary figures and tables.**
(PDF)

## Acknowledgments

We would like to thank Esther Rodman for her helpful discussions on cancer drug mechanisms of action. We thank members of the IMPROVE project for allowing us to join in on regular meetings and helping us to build the knowledge base necessary for this work. We would also like to thank all members of the Walther-Antonio laboratory for their feedback during project development and writing.

## Author contributions

**Conceptualization:** William G. Herbert, Nicholas Chia, Paul A. Jensen, Marina R.S. Walther-Antonio.

**Data curation:** William G. Herbert.

**Formal analysis:** William G. Herbert.

**Funding acquisition:** Marina R.S. Walther-Antonio.

**Investigation:** William G. Herbert.

**Methodology:** William G. Herbert, Paul A. Jensen.

**Resources:** William G. Herbert.

**Software:** William G. Herbert.

**Supervision:** Nicholas Chia, Paul A. Jensen, Marina R.S. Walther-Antonio.

**Validation:** William G. Herbert.

**Visualization:** William G. Herbert.

**Writing – original draft:** William G. Herbert.

**Writing – review & editing:** William G. Herbert, Nicholas Chia, Paul A. Jensen, Marina R.S. Walther-Antonio.

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
