## [Decision Letter · Decision Letter 0]

9 Aug 2025

Monotherapy cancer drug-blind response prediction is limited to intraclass generalization

PLOS Computational Biology

Dear Dr. Herbert,

Thank you for submitting your manuscript to PLOS Computational Biology. After careful consideration, we feel that it has merit but does not fully meet PLOS Computational Biology's publication criteria as it currently stands. Therefore, we invite you to submit a revised version of the manuscript that addresses the points raised during the review process.

Please submit your revised manuscript within 60 days Oct 09 2025 11:59PM. If you will need more time than this to complete your revisions, please reply to this message or contact the journal office at ploscompbiol@plos.org. Please include the following items when submitting your revised manuscript:

We look forward to receiving your revised manuscript.

Kind regards,

Hailin Chen

Academic Editor

PLOS Computational Biology

Pedro Mendes

Section Editor

PLOS Computational Biology

**Journal Requirements:**

3) Please amend your detailed Financial Disclosure statement. This is published with the article. It must therefore be completed in full sentences and contain the exact wording you wish to be published.

4) Your current Financial Disclosure states, "Yes ↳ Please add funding details. Funding of doctoral training for WGH provided by the Mayo Foundation for Medical Education and Research. ↳ Please select the country of your main research funder (please select carefully as in some cases this is used in fee calculation). UNITED STATES - US".

However, your funding information on the submission form indicates No funds received .

Please indicate by return email the full and correct funding information for your study and confirm the order in which funding contributions should appear. Please be sure to indicate whether the funders played any role in the study design, data collection and analysis, decision to publish, or preparation of the manuscript.

5) Kindly revise your competing statement to align with the journal's style guidelines: 'The authors declare that there are no competing interests.'

**Reviewers' comments:**

Reviewer's Responses to Questions

**Comments to the Authors:**

Reviewer #1: This manuscript provides a thorough and insightful analysis of drug-blind generalization limitations in cancer drug response prediction models. The authors systematically dissect the problem, shifting the focus from model architecture towards the intrinsic characteristics of datasets themselves. Key contributions include a compelling permutation experiment, which reveals that DRP models primarily learn drug-specific response distributions rather than detailed drug-cell line interactions. The authors also introduce a novel quantitative analysis demonstrating that generalizability in drug-blind scenarios is largely confined to drugs sharing the same or related mechanisms of action. The work persuasively argues that current failures in drug-blind prediction primarily reflect limitations in dataset structure rather than model capability. While the work presents a robust argument, the conclusions could be further strengthened by addressing several points:

1. The current experimental validation relies exclusively on MLPs. Although the simplicity of this approach effectively isolates data-driven effects, it remains uncertain if these insights hold for more advanced architectures such as Graph Neural Networks or Transformers. Conducting an architectural ablation study, particularly on critical experiments such as permutation experiments or MoA-specific training, would significantly enhance the claim of architectural independence. Alternatively, referencing these results more explicitly from existing literature could serve a similar purpose.

2. The intra-drug shuffle experiment convincingly shows models' biases towards learning average drug responses. Would it be plausible to conduct a complementary experiment where each drug's response values are replaced by a single representative value (like mean)? If the results are still similar to intra-drug shuffle, it would further confirm that the models are learning simplified drug-level heuristics.

3. The analysis comparing mechanism-specific expert models to a global model effectively highlights performance gains from focused training. However, it is unclear whether improvements result from genuine specialization or from avoiding "negative transfer"/noise from unrelated drug classes in global training. This could be clarified by introducing another control, where a model is trained on the union of selected mechanism-specific subsets, but excluding unrelated drug classes.

4. The paper does an excellent job of revealing the fundamental limitations of current drug-blind evaluation paradigms, but it stops short of proposing actionable alternatives. The research community would benefit from clearly defined recommendations or a new standardized evaluation framework to guide future benchmarking efforts.

Reviewer #2: The submitted manuscript presents an analysis of the limitations of drug response prediction (DRP) models, particularly their performance when applied to unseen compounds. The authors demonstrate that the observed performance in scenarios where drugs are already known to the model is largely driven by overlapping drug mechanisms, rather than by true generalization. This is indeed an interesting research topic; however, there exist several concerns which need to be addressed.

Major comments

1. The manuscript lacks readability in several places. Many sentences are overly long and complex, which affects the overall flow of the text. I recommend a thorough revision to improve clarity and ensure the manuscript reads more smoothly.

2. The rationale behind using a linear model with drug features as input is not clearly explained. The implementation details are also vague. For example, on page 19, line 125, the authors initially mention using a linear model, but only in the following sentence is it clarified that the model is in fact a regularized linear model (elastic net). This key detail should be stated up front, and the motivation for using this approach should be better explained.

3. The message conveyed by Figure 4 is unclear. Additionally, Figure 4a is difficult to read due to its visual complexity.

4. The choice of deep learning architecture is not clearly justified. In the model shown in Figure 8, the hyperparameter choices appear either arbitrary or specifically tuned to enhance test set performance. Specifically, the model uses a 7-layer architecture for the cell line wing and a 4-layer architecture for the drug wing, with custom neuron counts per layer. Although the choice of all these hyperparameters need to be justified, the manuscript states on page 17, line 353, that “the primary focus of this work was not to spend time identifying a hyperparameter tuning strategy.” This is concerning, as hyperparameter tuning is a fundamental step in machine learning. Even simpler models such as the elastic net, with only two hyperparameters, require a tuning via nested cross-validation, which is also absent in this manuscript.

5. The use of a deep neural network with ten layers is difficult to reconcile with the study’s focus on overfitting. Given the relatively limited number of samples, such a complex model increases the likelihood of overfitting, potentially undermining the main message of the paper. A more appropriate approach would have been to evaluate a range of simpler models to better support the analysis.

Miner comments

6. Whenever a Python package is mentioned, its version should also be provided to ensure reproducibility.

7. On page 8 line 133, there seems to be a redundant “be”: “Therefore, drugs are be,”.

8. The GitHub repository needs to be well documented and organized better.

Reviewer #3: Overall, the authors present an exciting and interesting study about the limitations of predicting drug responses in the context of limited compound diversity. The authors provide thoughtful interpretation of their analyses and effective case studies that highlight the impact of drug-class on prediction. I’m recommending a major revision because I think the methods lack clarity in some places and there’s an opportunity to better quantify drug compound similarity, generally, as well as defining drug-compound “composition” throughout.

Major:

- The GitHub respository is not public and so I cannot verify that the code and data are available.

- Generally, I think there should be more detail about the datasets used to better underscore later results. Speficially, the following details would be helpful:

o In the results, it’s a big jump to the permutation testing. What dataset was used and can you give a rough sense of scale of how many drug-cell line combinations were considered and how many drug repeats were used? It’s hard to understand how you did intra-drug shuffling without know how many or the nature (same dose?) of repeat drug samples.

o Similarly in the methods, is there a count of all drugs obtained and their replicates, if available? Also, is there a way to a priori characterize their mechanisms of action since the introduction so heavily emphasizes the importance of this information? Was there any overlap in the drugs or cell lines? And how were the response data harmonized across the three sources? Or did they all report the same response?

o Given the differences in the dataset size, prediction may be influenced by the response score and response score distribution. Could the authors give any insights about the differences in the response variables?

o For gCSI, can you give a range of cell lines per drug?

o For the tuples, how many features were extracted for cell lines? What is the intersection size with the TCGA data? Again, it’s likely sufficient to give a range. Also were all cell lines from the three DRP datasets available in TCGA?

o

- Could the authors better define dataset diversity? If the mechanism of action is so important, is there a way to first quantify the “diversity” of MoA? Perhaps even by clustering all of the fingerprints to show that each dataset has X unique groups? And then we selecting the test set, how was the “uniqueness” of the test set determined? This would be different than clustering drugs with annotated MOAs like in Figure 6.

- Could the authors give a more concrete definition of composition in the “Prediction of performance by training set composition” as well?

- In the results and methods section, can the authors provide more detail about how they did the one-hot encoding?

- I very much appreciate the discussion of the shared coefficients from Figure 4 – it’s interesting when drug targets in the same pathway can inform the model’s ability to predict unseen drugs. It would be helpful to describe how this information was obtained? The Figure 4 caption suggests that it was provided by GDSC, but the authors only mention drug fingerprints in the method and give no indication of retrieving their MOA or how these annotations were discovered.

- In the results section it would be helpful to remind the reader which drug response datasets are being used for each analysis, knowing that not all datasets had sufficient information for each analysis.

-

Minor:

- I wonder if the authors would consider highlighting a few concrete findings from these papers into these introduction sentences, which are intriguing: Early work examined the general scaling laws applicable to these datasets [8]. This is important for guiding further collection and experimentation. Different datasets also display different levels of efficacy for prediction among one another, showing that not all cancer DRP datasets are created equal.” What was learned about the general scaling? Was there a quantification of the extent of prediction from one dataset to another? These would be good numbers to contextualize the authors results in figure 6 & 7.

- “In one of the only efforts to directly address drug-blind failure, performance was improved using multi-objective optimization to prevent training from skewing performance to specific drugs by enhancing the extraction of global drug response features” Could the authors more concretely describe “global drug response” features? This comes up later and would be good to define.

- Does “mechanism of action” always refer to documented drug targets or something else? This is largely discussed for GDSC data, but do the other datasets contain any MOA information?

- In figure 8, the caption doesn’t sufficiently describe why there are multiple rectangles with decreasing numbers. Is this meant to show differences in features per cell line or that there is some prioritization of features before predicting the response?

- Please define all x-axis terms in the Figure 1 caption.

- The relatively high performance even with a low number of input samples is interesting, and has been shown in other places in the literature as well. I wonder if this paper, specifically figure 4C would be of interest to the authors. Dataset size is a very important topic for ML.

o Yuan B, Shen C, Luna A, Korkut A, Marks DS, Ingraham J, Sander C. CellBox: Interpretable Machine Learning for Perturbation Biology with Application to the Design of Cancer Combination Therapy. Cell Syst. 2021 Feb 17;12(2):128-140.e4. doi: 10.1016/j.cels.2020.11.013. Epub 2020 Dec 28. PMID: 33373583.

- For the elastic net in Figure 3B, is there any way to also plot the “predicted” or “real” score against some metric of training set drug “uniqueness”?

- A minor typo, “Therefore, drugs are be drug-blind”

- It would be helpful to make font sizes on figures larger throughout. Especially Figure 6 and 7.

- The relationship between PD0325901and PD173074 is interesting – I wonder if the authors may consider that their approach could uncover unknown drug targets when predictive utility is found between two compounds?

- Figure 6 is interesting in that it concretely identifies similarity in terms of reported MOA. Could you describe this data in the methods section- were all compounds annotated this way? In the results it says “selected compounds” but how many is that?

- Could the authors consider why some drug groups in Figure 6 remain disparate even after model training? Specifically, the PI3K drugs (blue dots) are disperse in Figure 6A, but even at epoch 100, they aren’t all cleanly grouped together, but seem to form distinct clusters within this mechanism of action. Again, this might highlight that even if drugs share at least one “primary target” that they could have distinct secondary or “off-targets”.

- Generally, the discussion would benefit from a first paragraph that summarizes the authors goals and findings before jumping to the conclusion that models only predict MOA.

- Is next-in-class prediction entirely predictable for drug-blind tasks? The authors show that tis depends greatly on the drug MOA as some classes, this is less true. Similarly, the authors also previously alluded to the fact that drugs with similar MOAs did not always have high fingerprint similarity. The last paragraph of the discussion is the most interesting in the context of this study’s results, but seemingly contradicts this earlier paragraph.

- I think the authors would benefit from a conclusion section that emphasizes their core findings and discoveries, many of which are very interesting, especially related to the diversity of compounds in DRP datasets and the impact of this information on prediction, instead of the speculatory content provided.

**Have the authors made all data and (if applicable) computational code underlying the findings in their manuscript fully available?**

The PLOS Data policy requires authors to make all data and code underlying the findings described in their manuscript fully available without restriction, with rare exception (please refer to the Data Availability Statement in the manuscript PDF file). The data and code should be provided as part of the manuscript or its supporting information, or deposited to a public repository. For example, in addition to summary statistics, the data points behind means, medians and variance measures should be available. If there are restrictions on publicly sharing data or code —e.g. participant privacy or use of data from a third party—those must be specified.requires authors to make all data and code underlying the findings described in their manuscript fully available without restriction, with rare exception (please refer to the Data Availability Statement in the manuscript PDF file). The data and code should be provided as part of the manuscript or its supporting information, or deposited to a public repository. For example, in addition to summary statistics, the data points behind means, medians and variance measures should be available. If there are restrictions on publicly sharing data or code —e.g. participant privacy or use of data from a third party—those must be specified.requires authors to make all data and code underlying the findings described in their manuscript fully available without restriction, with rare exception (please refer to the Data Availability Statement in the manuscript PDF file). The data and code should be provided as part of the manuscript or its supporting information, or deposited to a public repository. For example, in addition to summary statistics, the data points behind means, medians and variance measures should be available. If there are restrictions on publicly sharing data or code —e.g. participant privacy or use of data from a third party—those must be specified.requires authors to make all data and code underlying the findings described in their manuscript fully available without restriction, with rare exception (please refer to the Data Availability Statement in the manuscript PDF file). The data and code should be provided as part of the manuscript or its supporting information, or deposited to a public repository. For example, in addition to summary statistics, the data points behind means, medians and variance measures should be available. If there are restrictions on publicly sharing data or code —e.g. participant privacy or use of data from a third party—those must be specified.

Reviewer #1: Yes

Reviewer #2: Yes

Reviewer #3: **No:** Their GitHub link is not accessible. Perhaps it is private.Their GitHub link is not accessible. Perhaps it is private.Their GitHub link is not accessible. Perhaps it is private.Their GitHub link is not accessible. Perhaps it is private.

PLOS authors have the option to publish the peer review history of their article (what does this mean?). If published, this will include your full peer review and any attached files.). If published, this will include your full peer review and any attached files.). If published, this will include your full peer review and any attached files.). If published, this will include your full peer review and any attached files.

...

Reviewer #1: No

Reviewer #2: No

Reviewer #3: No

**Figure resubmission:**

**Reproducibility:**



---

## [Decision Letter · Decision Letter 1]

28 Dec 2025

Monotherapy cancer drug-blind response prediction is limited to intraclass generalization

PLOS Computational Biology

Dear Dr. Herbert,

Thank you for submitting your manuscript to PLOS Computational Biology. After careful consideration, we feel that it has merit but does not fully meet PLOS Computational Biology's publication criteria as it currently stands. Therefore, we invite you to submit a revised version of the manuscript that addresses the points raised during the review process.

We look forward to receiving your revised manuscript.

Kind regards,

Hailin Chen

Academic Editor

PLOS Computational Biology

Pedro Mendes

Section Editor

PLOS Computational Biology

**Additional Editor Comments:**

Please try to address the remaining comments from reviewer #2

**Journal Requirements:**

2) We have noticed that you have uploaded Supporting Information files, but you have not included a list of legends. Please add a full list of legends for your Supporting Information files after the references list.

**Reviewers' comments:**

Reviewer's Responses to Questions

**Comments to the Authors:**

Reviewer #1: The authors have addressed all of my previous comments with additional experiments and concrete reasoning. I appreciate their hard work and have no remaining substantive concerns.

Reviewer #2: Thank you for the response and for the additional experiments. I appreciate the inclusion of more complex architectures and I agree that the observation of qualitatively similar drug-blind behavior across these models strengthens the central claim that drug-blind failure is not specific to a single neural architecture.

That said, I believe some aspects of my original concern remain only partially addressed:

1. While the additional experiments with GraphCDR and scFoundation support the argument that increasing architectural complexity does not resolve drug-blind generalization, it is still not fully established that the observed failure mode is independent of model architecture. In addition, the manuscript still does not demonstrate whether simpler models exhibit the same qualitative behaviors that underpin the paper’s conclusions. The motivation for requesting simpler baselines was not to improve performance, but to more cleanly diagnose the nature of the learnable signal.

2. The clarification that the dominant form of overfitting studied here is data-driven is well taken and is supported by the permutation analyses. However, the strength of the manuscript’s conclusions — particularly statements suggesting that drug-blind performance reflects dataset behavior rather than model behavior — would be better justified by showing that these phenomena persist across a broader spectrum of model capacity and drug features.

3. Regarding the justification of the two-arm MLP architecture, I appreciate the addition of references [10, 11]. Having reviewed these works, I agree that they employ conceptually similar two-branch designs for drug response prediction. However, the architectures are not identical in depth, width, or layer-wise configuration, and the specific asymmetry adopted here (e.g., a deeper cell-line arm with custom layer sizes) remains only loosely motivated by precedent. Given the manuscript’s explicit data-centric framing, the absence of architectural sensitivity or ablation analyses leaves some uncertainty as to how much these design choices influence the observed behaviors.

4. While I appreciate the clarification that batch size and learning rate were selected via grid search, these parameters primarily affect optimization process rather than architectural capacity or inductive bias. As such, they do not directly address concerns related to model complexity, and representational power. The core architectural hyperparameters (e.g., number of layers, layer widths, and asymmetry between model arms) remain fixed and largely unexplored. The distinction between optimization hyperparameters and architectural hyperparameters is important.

5. I appreciate the authors’ effort to add a more detailed README to the GitHub repository. I encourage the authors to further improve reproducibility by clearly structuring the codebase and clarifying the recommended execution order. In addition, documenting the computational environment (e.g., Python version and package dependencies via a `requirements.txt` or conda environment file) would help ensure the results can be reliably reproduced.

In summary, while the evidence convincingly shows that current monotherapy pharmacogenomic datasets impose severe limits on drug-blind generalization, the stronger claim that this failure should be attributed primarily to data rather than model choice extends beyond what is directly demonstrated. If the authors do not intend to evaluate a substantially broader range of methods and feature representations—which would likely be outside the scope of the present work—then softening this claim, or more explicitly delimiting it to the architectures and settings used in the current study, would better align the conclusions with the evidence presented.

Reviewer #3: Thank you for your careful consideration of my comments and further explanation of the details and choices in your work.

**Have the authors made all data and (if applicable) computational code underlying the findings in their manuscript fully available?**

The PLOS Data policy requires authors to make all data and code underlying the findings described in their manuscript fully available without restriction, with rare exception (please refer to the Data Availability Statement in the manuscript PDF file). The data and code should be provided as part of the manuscript or its supporting information, or deposited to a public repository. For example, in addition to summary statistics, the data points behind means, medians and variance measures should be available. If there are restrictions on publicly sharing data or code —e.g. participant privacy or use of data from a third party—those must be specified.requires authors to make all data and code underlying the findings described in their manuscript fully available without restriction, with rare exception (please refer to the Data Availability Statement in the manuscript PDF file). The data and code should be provided as part of the manuscript or its supporting information, or deposited to a public repository. For example, in addition to summary statistics, the data points behind means, medians and variance measures should be available. If there are restrictions on publicly sharing data or code —e.g. participant privacy or use of data from a third party—those must be specified.requires authors to make all data and code underlying the findings described in their manuscript fully available without restriction, with rare exception (please refer to the Data Availability Statement in the manuscript PDF file). The data and code should be provided as part of the manuscript or its supporting information, or deposited to a public repository. For example, in addition to summary statistics, the data points behind means, medians and variance measures should be available. If there are restrictions on publicly sharing data or code —e.g. participant privacy or use of data from a third party—those must be specified.requires authors to make all data and code underlying the findings described in their manuscript fully available without restriction, with rare exception (please refer to the Data Availability Statement in the manuscript PDF file). The data and code should be provided as part of the manuscript or its supporting information, or deposited to a public repository. For example, in addition to summary statistics, the data points behind means, medians and variance measures should be available. If there are restrictions on publicly sharing data or code —e.g. participant privacy or use of data from a third party—those must be specified.

Reviewer #1: Yes

Reviewer #2: Yes

Reviewer #3: Yes

PLOS authors have the option to publish the peer review history of their article (what does this mean?). If published, this will include your full peer review and any attached files.). If published, this will include your full peer review and any attached files.). If published, this will include your full peer review and any attached files.). If published, this will include your full peer review and any attached files.

...

Reviewer #1: No

Reviewer #2: No

Reviewer #3: No

**Figure resubmission:**
---

## [Decision Letter · Decision Letter 2]

26 Mar 2026

Dear Dr. Herbert,

We are pleased to inform you that your manuscript 'Monotherapy cancer drug-blind response prediction is limited to intraclass generalization' has been provisionally accepted for publication in PLOS Computational Biology.

Best regards,

Hailin Chen

Academic Editor

PLOS Computational Biology

Pedro Mendes

Section Editor

PLOS Computational Biology

This manuscript can be accepted for publication. Meanwhile, please make some revision according to the Reviewer's suggestion.

Reviewer's Responses to Questions

**Comments to the Authors:**

Reviewer #2: The submitted revision shows an improved version of the manuscript, in particular in the effort to clarify and appropriately scope the conclusions. The added analysis of architectural hyperparameters is useful. Interestingly, performance appears largely unchanged across a wide range of model configurations. This observation also relates to my initial comment regarding the justification of the model architecture, as it suggests that the selected architecture does not provide a clear advantage over simpler alternatives (e.g., a single-layer perceptron). At the same time, this finding is consistent with the authors' overall argument and directly supports their conclusions. Therefore, I encourage the authors to incorporate this analysis in the main text rather than limiting it to the reviewer response. Overall, the revisions are satisfactory, and acceptance is recommended.

**Have the authors made all data and (if applicable) computational code underlying the findings in their manuscript fully available?**

The PLOS Data policy requires authors to make all data and code underlying the findings described in their manuscript fully available without restriction, with rare exception (please refer to the Data Availability Statement in the manuscript PDF file). The data and code should be provided as part of the manuscript or its supporting information, or deposited to a public repository. For example, in addition to summary statistics, the data points behind means, medians and variance measures should be available. If there are restrictions on publicly sharing data or code —e.g. participant privacy or use of data from a third party—those must be specified.requires authors to make all data and code underlying the findings described in their manuscript fully available without restriction, with rare exception (please refer to the Data Availability Statement in the manuscript PDF file). The data and code should be provided as part of the manuscript or its supporting information, or deposited to a public repository. For example, in addition to summary statistics, the data points behind means, medians and variance measures should be available. If there are restrictions on publicly sharing data or code —e.g. participant privacy or use of data from a third party—those must be specified.requires authors to make all data and code underlying the findings described in their manuscript fully available without restriction, with rare exception (please refer to the Data Availability Statement in the manuscript PDF file). The data and code should be provided as part of the manuscript or its supporting information, or deposited to a public repository. For example, in addition to summary statistics, the data points behind means, medians and variance measures should be available. If there are restrictions on publicly sharing data or code —e.g. participant privacy or use of data from a third party—those must be specified.requires authors to make all data and code underlying the findings described in their manuscript fully available without restriction, with rare exception (please refer to the Data Availability Statement in the manuscript PDF file). The data and code should be provided as part of the manuscript or its supporting information, or deposited to a public repository. For example, in addition to summary statistics, the data points behind means, medians and variance measures should be available. If there are restrictions on publicly sharing data or code —e.g. participant privacy or use of data from a third party—those must be specified.

Reviewer #2: Yes

PLOS authors have the option to publish the peer review history of their article (what does this mean?). If published, this will include your full peer review and any attached files.). If published, this will include your full peer review and any attached files.). If published, this will include your full peer review and any attached files.). If published, this will include your full peer review and any attached files.

...

Reviewer #2: No

---

## [Editor Report · Acceptance letter]

PCOMPBIOL-D-25-01185R2

Monotherapy cancer drug-blind response prediction is limited to intraclass generalization

Dear Dr Herbert,

I am pleased to inform you that your manuscript has been formally accepted for publication in PLOS Computational Biology. Your manuscript is now with our production department and you will be notified of the publication date in due course.

With kind regards,

Judit Kozma
